



**Intercomparison between the Integrated Urban land Model and the**
**Noah Urban Canopy Model**
Chunlei MENG    Junxia DOU
Institute of Urban Meteorology, China Meteorological Administration, 100089,
Beijing, China.
Corresponding author: Dr. Chunlei Meng, Institute of Urban Meteorology, China
Meteorological Administration, Beijing, 100089, China. No.55 Beiwaxili, Haidian
District, Beijing, China. E-mail: clmeng@ium.cn
Mobile:(86)13811346500





**Abstract**
Urban land surface model (ULSM) is an important tool to study the climatic effect
of human activity. Now there are two main methods to parameterize the effects of
human activity, the coupling method and the integrating method. For the coupled
method, the urban canopy model (UCM) was developed and coupled with the land
surface model for the natural land surfaces. For the integrated method, the urban
land surface model was built directly based on the traditional land surface model. In
this paper, the Noah Single Layer Urban Canopy Model (Noah/SLUCM) and the
Integrated Urban land Model (IUM) were compared using the observed fluxes data at
the 325-meter meteorology tower in Beijing. Through the comparison, the key
factors and physical processes of the urban land surface model which have significant
impact on the performance of ULSM were found out. The results indicate that the
absorbed solar radiation of urban surface was reduced by the solar radiation
scattering, the absorption of building roof and wall, and the shading effect of urban
canopy and tall buildings. Urban surface roughness length and friction velocity are
important in urban sensible heat flux simulation. Urban water balance and
impervious surface evaporation (ISE) are important in urban latent heat flux
simulation.
**Key words**: integrated urban land model; urban canopy model; reduction coefficient
of solar absorption; impervious surface evaporation; urban water balance





## 1. Introduction


China now is experiencing an unprecedented urbanization movement. The
regional climate could be changed by urbanization through the greenhouse gases
emission, land use and land cover change (LUCC), and anthropogenic heat release
(AHR) (Committee on Urban Meteorology 2012). The impact of land use and land
cover change (LUCC) on climate change has the same order of magnitude as the
impact of greenhouse gases emissions (Kalnay and Cai 2003; Sun et al. 2016). Urban
land surface model is an important tool to study the climatic effect of urbanization.
Now there are two main methods to parameterize the effects of human activity. One
is the coupled method and the other is the bulk or integrated method. For the
coupled method, the urban canopy model (UCM) (e.g. Kusaka et al. 2001; Kusaka and
Kimura 2004) was developed and coupled with the land surface model for the natural
land surfaces. For the integrated method (e.g. Meng 2015), the urban land surface
model was built directly based on the traditional land surface model.
Model intercomparison is an effective method to evaluate different land surface
models. Through the comparison, the key factors of the urban land surface model
were found out. In the 1990s, the Project for Intercomparison of Land Surface
Parameterization Schemes (PILPS) (Henderson-Sellers and Brown 1992;
Henderson-Sellers et al. 1993, 1995) was launched to compare land surface models.
Along the development of the urban land surface parameterization schemes, urban
land surface model comparison project (Grimmond et al. 2010, 2011; Best and
Grimmond 2015) was launched. The results indicate that the urban database (Ching
et al. 2009; Demuzere et al. 2017; Hammerberg et al. 2018; He et al. 2015; Loridan
and Grimmond 2012) is very important to urban land surface models. Compared with



the parameterization of the urban energy balance processes, recently developed
urban land surface models could not parameterize the urban water balance
(Grimmond et al. 1986; Mitchell et al. 2001; Wang et al. 2013; Miao and Chen 2014;
Yang et al. 2015) processes accurately. Albedo and fractional vegetation cover are
important parameters in urban land surface model, while the AHR is only important
in certain cities.
However, these intercomparison studies only concern the impact of the bulk
model and UCM with different complexity on surface radiation and energy balances.
In this paper, different from the models used for intercomparison in Best and
Grimmond (2015), the Integrated Urban land Model (IUM) (Meng 2015) was
introduced for intercomparision. In IUM, for urban underlying surfaces, a water
balance model was developed using the impervious water depth as the prognostic
variable. The Noah Single Layer Urban Canopy Model (Noah/SLUCM) (Chen et al.
2011) and the IUM (Meng 2015) were intercompared using the observed fluxes data
at 325-meter Beijing meteorology tower in Beijing. In addition to the comparison of
the fluxes in the radiation and energy balance equations, the ISE in waterlogging
circumstance was also compared to study the mechanism of urban water balance
processes. Through the comparison, the key factors and physical processes in the
urban land surface model were found out.

## 2. Model and Data

### 2.1 Model

Three main control equations exist in the land surface model; they are radiation
balance model, energy balance model and water balance model. We compared all



the simulated fluxes in the radiation and energy balance equations with the
observation. The radiation balance model could be described as follows:
$$R_n = S\downarrow + L\downarrow - S\uparrow - L\uparrow \qquad (1)$$
Where $R_n$ is the net radiation (W m$^{-2}$); $S\downarrow$ is the downward solar radiation;
$L\downarrow$ is the downward longwave radiation (W m$^{-2}$); $S\uparrow$ is the upward solar
radiation (W m$^{-2}$); $L\uparrow$ is the upward longwave radiation (W m$^{-2}$).
The energy balance model could be described as follows:
$$R_n = H + LE + G + A \qquad (2)$$
Where $H$ is the sensible heat flux (W m$^{-2}$); $L$ is the latent heat of evaporation
for water (W m$^{-2}$); $E$ is the evapotranspiration (W m$^{-2}$); $LE$ is the latent heat flux
(W m$^{-2}$); $G$ is the ground heat flux (W m$^{-2}$); $A$ is the AHR (W m$^{-2}$), which used the
same diurnal cycle data for the two models.
As this paper focuses on the fluxes in urban areas, only the parameterization
schemes of the fluxes in urban areas is given blow. The detailed parameterization
scheme of the models could be seen in relevant papers (Chen and Dudhia, 2001;
Meng, 2015).
*2.1.1 Noah/SLUCM*
The Noah Land Surface Model (Noah LSM) (Chen et al. 1996; Chen and Dudhia
2001; Ek et al. 2003) has been implemented in the fifth-generation Pennsylvania
State University–NCAR Mesoscale Model (MM5) and the weather research and
forecasting (WRF) model. It was developed by National Center for Atmospheric
Research (NCAR), National Centers for Environmental Prediction (NCEP), the U.S. Air
Force Weather Agency (AFWA), and the university community. The single-layer urban





canopy model (SLUCM) was developed by Kusaka *et al*. (2001) and Kusaka and
Kimura (2004). It assumes infinitely-long street canyons parameterized to represent
urban geometry, but recognizes the three-dimensional nature of urban surfaces. The
SLUCM has been coupled with the Noah LSM in the community mesoscale WRF
Model, version 2.2 (Chen et al. 2011). The fluxes in Noah/SLUCM are divided into
three parts; they are fluxes in roof, wall and road respectively.
The upward shortwave radiation is parameterized as follows:
$$S\uparrow = S\downarrow - \left[ RS_R + 2H_{GT}S_B + (1-R)S_G \right] \qquad (3)$$

Where $S\uparrow$ is the upward shortwave radiation (W m$^{-2}$); $S\downarrow$ is the downward
shortwave radiation (W m$^{-2}$); $R$ is the normalized roof width; $H_{GT}$ is the
normalized building height; $S_R$, $S_B$ and $S_G$ are the absorbed solar radiation by
roof, wall and road respectively (W m$^{-2}$).
The upward longwave radiation is parameterized as follows:
$$L\uparrow = L\downarrow - \left[ RR_R + 2H_{GT}R_B + (1-R)R_G \right] \qquad (4)$$

Where $L\uparrow$ is the upward longwave radiation (W m$^{-2}$); $L\downarrow$ is the downward
longwave radiation (W m$^{-2}$); $R_R$, $R_B$ and $R_G$ are the absorbed longwave radiation
by roof, wall and road respectively (W m$^{-2}$).
The sensible heat flux is parameterized as follows:
$$H = RH_R + 2H_{GT}H_B + (1-R)H_G \qquad (5)$$

Where $H$ is the sensible heat flux (W m$^{-2}$); $H_R$, $H_B$ and $H_G$ are the sensible
heat flux from roof, wall and road respectively (W m$^{-2}$).
The ISE is parameterized as follows:
$$E_{imp} = RE_R + 2H_{GT}E_B + (1-R)E_G \qquad (6)$$





141  Where $E_{imp}$ is the ISE (mm s$^{-1}$); $E_R$, $E_B$ and $E_G$ are the evaporation from roof,

142  wall and road respectively (mm s$^{-1}$), they are calculated as follows:

$$E_R = Ch_r u_a \beta_r \left( q_{sr} - q_a \right) \tag{7}$$

$$E_B = Ch_b u_c \beta_b \left( q_{sb} - q_c \right) \tag{8}$$

$$E_G = Ch_g u_c \beta_g \left( q_{sg} - q_c \right) \tag{9}$$

146  Where $Ch_r$, $Ch_b$ and $Ch_g$ are the heat transfer coefficient from roof, wall and

147  road respectively (mm m$^{-1}$); $u_a$ and $u_c$ are wind speed in the reference height and

148  canopy respectively (m s$^{-1}$); $\beta_r$, $\beta_b$ and $\beta_g$ are the evaporation coefficients that

149  are regulated by the availability of moisture for roof, wall and road respectively; they

150  could be calculated as follows (Miao and Chen, 2014):

151
$$\beta_g = \begin{cases} \begin{cases} 1.0, & Rain \ge 10mm, \\ 0.5, & 0 < Rain < 10mm, \end{cases} & Rain, \\ \beta_{g,0} e^{(-dt/5)}, & No\ rain, \end{cases} \tag{10}$$

152
$$\beta_r = \beta_b = \beta_g$$

153  Where $Rain$ is the daily precipitation; $\beta_{g,0}$ is $\beta_g$ at the previous step; $dt$ is

154  the time step (h).

155  $q_{sr}$, $q_{sb}$ and $q_{sg}$ are the saturated specific humidity for roof, wall and road

156  respectively (kg kg$^{-1}$); $q_a$ and $q_c$ are the specific humidity for reference height and

157  canopy respectively (kg kg$^{-1}$).

158  The ground heat flux is parameterized as follows:

$$G = RG_R + 2H_{GT}G_W + (1 - R)G_G \tag{11}$$

160  Where $G$ is the surface heat flux (W m$^{-2}$); $G_R$, $G_B$ and $G_G$ are the surface

161  heat flux from roof, wall and road respectively (W m$^{-2}$).



*2.1.2 IUM*
The Integrated Urban land Model (IUM) (Meng 2015) was developed based on the
Common Land Model (CoLM) (Dai et al. 2003). IUM integrates the land surface
models for urban and natural land surfaces. For urban land surfaces, the energy
balance model was improved and the water balance model for impervious surfaces
was developed.
The upward shortwave radiation is parameterized as follows (Dai et al. 2003):
$$S\uparrow = S\downarrow \cdot \alpha \tag{12}$$
Where $\alpha$ is the albedo, which is defined as follows (Dai et al., 2003):
$$\alpha = \left(\alpha_{vis,dif} + \alpha_{nir,dif}\right)/2 \tag{13}$$
Where $\alpha_{vis,dif}$ and $\alpha_{nir,dif}$ are the albedo for visible and near infrared diffuse
solar radiation respectively, which are set as the same as the albedo of the saturated
soil with darkest color in CoLM, they are 0.05 and 0.1 respectively.
The upward longwave radiation is parameterized as follows:
$$L\uparrow = (1 - L\downarrow)\varepsilon + (1 - F_{cov})\varepsilon\sigma T_g^4 + F_{cov}\varepsilon\sigma T_l^4 \tag{14}$$
Where $\varepsilon$ is the emissivity; $\sigma$ is the Stefan-Boltzmann constant (W m$^{-2}$ K$^{-4}$);
$F_{cov}$ is the fractional vegetation cover; $T_g$ is the ground surface temperature (K);
$T_l$ is the leaf temperature (K).
The sensible heat flux is parameterized as follows:
$$H = \rho_a c_p \frac{\theta_g - \theta_a}{r_{ah}} = \rho_a c_p \frac{T_g - \theta_a}{r_{ah}} \tag{15}$$
Where $\rho_a$ is the air density (Kg m$^{-3}$); $c_p$ is the specific heat of dry air (J Kg$^{-1}$
K$^{-1}$); $\theta_g$ is the surface potential temperature (K); $\theta_a$ is the air potential



temperature at reference height (K); $r_{ah}$ is the aerodynamic resistance for sensible
heat flux between the atmosphere at reference height and the surface (m s$^{-1}$), which
could be calculated as follow:
$$r_{ah} = v u_* / f_h \tag{16}$$

Where $v$ is the von Karman constant; $u_*$ is the friction velocity (m s$^{-1}$); $f_h$ is
the integral of profile function for heat, which is associated with the thermodynamic
roughness.
The ISE in the IUM it is parameterized as follows:
$$E_{imp} = \begin{cases} E_p & (W_{i-1} > 0 \text{ and } W_i > 0) \\ \min\left(E_p, \max\left(0, \left(P_{rcp} - D_{rain}\right)\right)\right) & (W_{i-1} > 0 \text{ and } W_i = 0) \end{cases} \tag{17}$$

Where $E_p$ is the potential evaporation (m s$^{-1}$), $W_{i-1}$ is the road water depth
(mm) in the previous time step, and $W_i$ is the road water depth (mm) in the current
time step. The road water depth is controlled by the road water balance equation
which will be discussed in the next part of the paper. $E_p$ can be parameterized as
follows:
$$E_p = \frac{1000}{\rho_w} \rho_a \frac{q_{sat} - q_m}{r_d} \tag{18}$$

Where $\rho_a$ is the air density (kg m$^{-3}$); $\rho_w$ is the water density (kg m$^{-3}$), which is
approximately equal to 1000; $r_d$ is the aerodynamic resistance for evaporation (s
m$^{-1}$); $q_m$ is the specific humidity of the air (kg kg$^{-1}$); and $q_{sat}$ is the saturated
specific humidity of the water surface (kg kg$^{-1}$). If the water depth is not zero, road
surfaces are treated as shallow lakes, and the lake model (Henderson-Sellers 1986;
Hostetler and Bartlein 1990; Hostetler et al. 1993) in the CoLM is simplified to





compute the road water temperature. The simplified shallow lake model can be
described as follows:

$$\frac{\partial T_w}{\partial t} = \frac{\partial}{\partial W}\left[(\frac{k_w}{c_w} + k_e)\frac{\partial T_w}{\partial W}\right] + \frac{1}{c_w}\frac{d\phi}{dW}$$

(19)

Where $T_w$ is the road water temperature (K), $k_w$ is the thermal conductivity of
water (W m$^{-1}$ K$^{-1}$), $k_e$ is the eddy diffusion coefficient (m$^2$ s$^{-1}$), $c_w$ is the heat
capacity of water (J m$^{-3}$ K$^{-1}$), and $\phi$ is the solar radiation heat source term (W m$^{-2}$).
The ground heat flux is considered as the remainder of the energy balance
equation.
**2.2 Data**
The Chinese Academy of Sciences 325-m-high Meteorology and Environmental
Observation Tower data was used to force the models and for comparison. The
tower is located in downtown Beijing, the altitude of the foot of the tower is 49m,
the longitude and latitude are 116.3708E and 39.9744N respectively. The turbulent
heat fluxes including the sensible heat flux and latent heat flux are measured using
the eddy covariance (EC) technique at the 47-meter height. The radiation fluxes
including the upward and downward shortwave and longwave radiation are
measured using the radiometer at the 47-meter height. The net radiation was
calculated using equation 1. The ground heat flux was calculated using equation 2;
the AHR was not considered. The Gaofen-2 2m resolution LULC data (Figure 1) was
used to calculate the area percentage of each LULC categories. Within a 1000m
radius of the tower, the surface is 78.3% impervious (buildings, roads, etc) and 21.7%
pervious (trees and grass). The simulation time period is from 1st March to 31st
October 2015. The temporal resolution of the two models is 30min.





**2.3 Model Parameterization and Initialization**
Table 1 listed the sources of the initial values of the time-variant variables, forcing
variables, and the values of the time-invariant variables for the two models. In order
to ensure the objectivity of the intercomparison, identical values were set up for the
variables used both in Noah/SLUCM and IUM. For the variables only used in
Noah/SLUCM, the values were parameterized using the default look-up table. All
other variables referred by the paper in section 2.1 are intermediate variables, which
were calculated by the models themselves. The initial values of soil moisture, surface
temperature and soil temperature are from the observation of the tower. The LULC
categories are from Gaofen-2 data which referred in the last section. The diurnal
cycle of the anthropogenic heat is the same for the two models which is used from
Miao et al. (2011). The meteorological forcing data is from the observation of the
tower. The leaf area index (LAI) and fractional vegetation cover (FVC) are considered
as the forcing variables which calculated using an empirical equation (Dai et al. 2003;
Wang et al. 2007) as follows:
$$LAI = LAI_{max}(LULC) + \left(LAI_{min}(LULC) - LAI_{max}(LULC)\right)(1 - f) \qquad (20)$$
$$FVC = 1 - \exp\left(- K \cdot LAI\right) \qquad (21)$$
Where $LAI$ is the LAI; $LAI_{max}$ is the maximum LAI; $LAI_{min}$ is the minimum LAI;
$LULC$ is the LULC; $f$ is the coefficient to calculate LAI, which is associated with
the soil temperature in a certain soil depth, above this depth, 90 percent root
fraction was included; $FVC$ is the FVC; $K$ is the direct solar extinction coefficient,
which is associated with the leaf angle distribution factor which is designated
according to the LULC.




## 3. Results and Discussion

### 3.1 Radiation Balance

The upward shortwave radiation is determined by the albedo. The albedo is zero
at night. From the daily mean (Figure 2a) and diurnal cycle (Figure 2b) of upward
shortwave radiation, it is concluded that the simulation result of IUM is very good
compared with the observation especially from 9:00 to 16:00. The Albedo in
Noah/SLUCM is too large compared with the observation. Albedo is very important
in radiation balance, the remote sensing retrieved albedo (Xu and Shu 2014) should
be assimilated into land surface model in the future.
The upward longwave radiation (Figure 3) is associated with the emissivity and the
surface radiative temperature. The simulated upward longwave radiation is higher
than the observation in the daylight. Compared with that of the Noah/SLUCM, the
upward longwave radiation simulated by IUM has a greater deviation from 8:00 to
17:00, as the ground surface temperature simulated by IUM is too high during this
time period. In IUM, the solar radiation absorption of the ground surface is
overestimated as the shading effect of urban canopy and buildings and the multiple
scattering of the solar radiation are not considered (Wang et al. 2016). In order to
improve the simulation performance of IUM, the reduction coefficient of solar
radiation absorption which is associated with the urban canyon direction, building
height, road width ratio, sky view factor, and fractional vegetation cover etc. should
be considered. For Noah/SLUCM, the simulated upward longwave radiation is a little
higher than the observation at night; while for IUM, it is a litter lower than the





observation. Due to the complicated parameterization of the UCM, the Noah/SLUM
is superior to IUM in upward longwave radiation modeling.
Figure 4 shows the simulated net radiation compared with the observation. The
net radiation is also an important component in surface energy balance (Offerle et al.
2003). The observed net radiation is calculated by equation 1; it is associated with
the upward shortwave and longwave radiation. The net radiation is underestimated
by both the two models from 8:00 to 20:00. Compared with that of the Noah/SLUCM,
the IUM can simulate the net radiation better both in the daylight and at night. The
net radiation is considered as the remainder of the surface radiation balance
equation; the simulation results are depended on both the upward shortwave and
longwave radiation.
**3.2 Energy Balance**
The sensible heat flux is associated with the ground surface temperature and the
heat transfer resistance. The heat transfer resistance is associated with the
thermodynamic roughness and the friction velocity (See eq. 15). For urban land
surface, the thermodynamic roughness and friction velocity are relatively larger than
those in the natural surface. The parameterization of friction velocity should be
studied in the future in order to improve the simulation of sensible heat flux. The
simulated sensible heat flux for Noah/SLUCM and IUM are apparently larger than
those of the observation in the daylight. Compared with that of the Noah/SLUCM,
the sensible heat flux simulated by IUM has a greater deviation, as the ground
surface temperature simulated by IUM is too high in the daylight especially from
9:00 to 17:00 (Figure 5). At night, the simulation of IUM is compared well with the
observation. Due to the complicated parameterization of the UCM, the Noah/SLUM





is superior to IUM in sensible heat flux modeling too.

The ISE parameterization schemes for Noah/SLUCM and IUM are different (See eq.

10 and 17). For Noah/SLUCM, it is associated with the precipitation; while for IUM, it
is associated with the surface water depth, precipitation and drainage. From figure 6,
it is concluded that the simulated ISE by IUM is apparently larger than that simulated
by Noah/SLUCM both in the daylight and at night; but it is still lower that of the
observation. One of the reasons is the accuracy of the impervious surface
percentage is questionable; the other reason is the drainage is hard to parameterize.
The characteristic scale of the observed fluxes is crucial important for urban
heterogeneous underlying surfaces. Except for these limitations, urban water
balance model is still important in the parameterization of ISE; this would be
discussed at length in the next section.

Figure 7 is the simulated ground heat fluxes compared with the observation. The

ground heat temperature is associated with the ground temperature and the soil
temperature in the second layer; it is considered as the remainder term of the
surface energy balance equation for observation. Noah/SLUCM has four soil layers;
while IUM has 10 layers. The ground heat flux is underestimated by all the two
models in the daylight from 8:00 to 18:00. Compared with that of the Noah/SLUCM,
the simulated by IUM has a greater deviation during this time period. As the
Noah/SLUCM is superior to IUM in the simulation of urban surface temperature, the
ground heat flux simulation of Noah/SLUCM is also better than that of IUM
compared with the observation. IUM should be coupled with SLUCM to improve the
simulation performance in upward longwave radiation, sensible and ground heat
flux.



### 3.3 Urban Water Balance

The ISE is an important parameter in urban water balance (Yang et al. 2015). The water depth in impervious surface is an important parameter in urban waterlogging study which is not included in the Noah/SLUCM. In IUM, the water depth in impervious surface is considered as the prognostic variable of urban balance equation. The urban water balance equation in IUM could be described as follows:

$$\frac{\partial W}{\partial t} = P_{rcp} - I_{roof} - E_{imp} - D_{rain} - P_{er}I_{nf} \tag{22}$$

Where $W$ is the water depth on impervious surface (mm); $I_{roof}$ is the roof rainfall interception (mm s$^{-1}$) (Nakayoshi et al, 2009); $P_{er}$ is the percentage of the pervious surface; and $I_{nf}$ is the infiltration rate (mm s$^{-1}$).

An empirical equation is developed in Noah/SLUCM to parameterize the ISE, it is only associated with the daily precipitation and the time step (Eq. 10). Figure 8 shows the observed precipitation and simulated water depth by IUM during the simulation period (Figure 8). The drying of the impervious surface should last a period of time. As the result, the impervious surface evaporation is not just associated with the daily precipitation. For IUM, the ISE is parameterized based on the physical mechanisms which control the urban water balance and urban water temperature (Eq. 17-19, 22).

Figure 9 is the observed and simulated diurnal cycle of the latent heat fluxes in waterlogging days. Apparently, the simulated ISE of Noah/SLUCM is smaller than that of the observation. ISE plays an important role in rainy days; the simulation of latent heat flux by IUM is increased compared with that of Noah/SLUCM and it is very close to the observation both in the daylight and at night. The urban water balance model





is indispensable in urban latent heat flux simulation and urban hydrology research.
**3.4 Quantitative Comparison**

Figure 10 is the scatted plots of the simulated upward shortwave radiation,

upward longwave radiation, net radiation, sensible heat flux, latent heat flux and
ground heat flux by the simulation of Noah/SLUCM and IUM compared with the
observation. Table 2 and figure 11 are the biases, mean errors (MEs), root mean
square errors (RMSEs) and correlation coefficients (Rs) of these fluxes simulated by
the two models compared with those of the observations. Compared with the
Noah/SLUCM, IUM simulated the upward shortwave radiation quite well. The bias,
ME and RMSE are 0.32W/m$^2$, 2.74 W/m$^2$ and 5.03 W/m$^2$ respectively. Both two
models overestimate the upward longwave radiation and the sensible heat flux; the
deviation of IUM is larger than that of Noah/SLUCM. Both models underestimate the
net radiation and the latent heat flux; the deviation of IUM is smaller than that of
Noah/SLUCM. Both models underestimate the ground heat flux, the deviation of IUM
is larger than that of Noah/SLUCM. Through the quantitative comparison, it is
concluded that the Noah/SLUCM can simulate the urban surface temperature,
sensible heat flux and upward longwave radiation well; while the IUM could simulate
the ISE more accurately. The simulation result of the upward shortwave radiation is
depended on the albedo.

**4. Conclusions**

In this paper, the Noah/SLUCM and IUM were intercompared using the observed

fluxes data at 325-meter meteorology tower in Beijing. Through the comparison, the
key factors and physical processes of the urban land surface model were found out.





The characteristic scale of the observed fluxes is crucial important for urban
heterogeneous underlying surface. The albedo is an important factor in surface
radiation partition. In order to improve the simulation accuracy, remote sensing
retrieved albedo should be assimilated into land surface model. The solar radiation
absorption of urban surface is reduced by the multiple scattering, absorption of
building roof and wall, and shading effect of urban canopy and tall buildings. The
reduction coefficient of solar radiation absorption is associated with the urban
canyon direction, building height, road width ratio, sky view factor, and fractional
vegetation cover etc. Urban thermodynamic roughness and friction velocity are
important in urban sensible heat flux simulation. Urban thermodynamic roughness is
associated with the height of urban canopy. Urban friction velocity is usually larger
than that in the rural area. The drying of the impervious surface should last a period
of time. So, the impervious surface evaporation is not just associated with the daily
precipitation. ISE is important in rainy days. The urban water balance model is
indispensable in urban latent heat flux simulation and urban hydrology research.
In the near future, the characteristic scale of the observed fluxes should be
studied by using the footprint method (Roth et. al, 2017). The World Urban Database
and Access Portal Tools (WUDAPT) (Hammerberg et al. 2018) should be used in IUM
to parameterize the reduction coefficient of solar radiation absorption. The MODIS
retrieved albedo should be assimilated into IUM. The urban thermodynamic
roughness and urban friction velocity should be reparameterized too.






**Code and Data Availability**

The supplement of the paper includes the data for the 325m tower and the code of the IUM and the Noah/SLUCM.

**Author Contributions**

C.M. conceived the study and wrote the initial draft of the paper. J.D. drew some the pictures and dealt with the paper. All authors revised the paper.

**Competing interests**

The authors declare that they have no competing interests.

**Acknowledgements**

This work was supported by the National Natural Science Foundation of China under Grant 41875125 and 41705086. We thank Institute of Atmospheric Physics (IAP), CAS for Beijing meteorology tower observation data; National Center for Atmospheric Research (NCAR) for Noah/SLUCM.



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





Table 1 The sources of the initial values of the time-variant variables, forcing variables, and the
values of the time-invariant variables for IUM and Noah/SLUCM.

| | Variables | IUM | Noah/SLUCM |
|---|---|---|---|
| Time-variant | Soil Moisture | Observation | Observation |
| | Surface temperature | Observation | Observation |
| | Soil temperature | Observation | Observation |
| | LULC categories | Gaofen-2 | Gaofen-2 |
| Forcing | Anthropogenic heat | Diurnal cycle | Diurnal cycle |
| | Meteorological forcing | Observation | Observation |
| | Leaf area index | Empirical equation | Empirical equation |
| | Fractional vegetation cover | Empirical equation | Empirical equation |
| Time-invariant | Emissivity of ground | 0.96 | 0.96 |
| | Emissivity of wall | / | 0.96 |
| | Emissivity of roof | / | 0.96 |
| | Roof level | / | Look-up table |
| | Roof width | / | Look-up table |
| | Heat capacity of roof | / | Look-up table |
| | Heat capacity of wall | / | Look-up table |
| | Thermal conductivity of roof | / | Look-up table |
| | Thermal conductivity of wall | / | Look-up table |
| | Heat capacity of impervious ground surface | $2.025 \times 10^6$ J m$^{-3}$K$^{-1}$ | $2.025 \times 10^6$ J m$^{-3}$K$^{-1}$ |
| | Thermal conductivity of impervious ground surface | 2.9 Wm$^{-1}$K$^{-1}$ | 2.9 Wm$^{-1}$K$^{-1}$ |
| | Albedo of roof | / | Look-up table (0.20) |
| | Albedo of wall | / | Look-up table (0.20) |
| | Albedo of ground | 0.075 | 0.075 |
| | Street direction | / | Look-up table |
| | Street width | / | Look-up table |
| | Building width | / | Look-up table |
| | Building height | / | Look-up table |
| | Number of roof layer | / | 4 |



| | | |
|---|---|---|
| Roof layer thickness | / | Look-up table |
| Number of wall layer | / | 4 |
| Wall layer thickness | / | Look-up table |
| Number of road layer | 4 | 4 |
| Road layer thickness (m) | 0.0071, 0.028, 0.062, 0.119 | 0.0071, 0.028, 0.062, 0.119 |
| Roughness length of roof (m) | / | 0.01 |
| Roughness length of wall (m) | / | 0.0001 |
| Roughness length of ground (m) | 0.01 | 0.01 |

























Table 2 Biases, mean errors (MEs), root mean square errors (RMSEs) and correlation
coefficients (Rs) of the fluxes simulated by the two models compared with those of the

observations.

| Variables and Models | | Biases(W/m$^2$) | MEs(W/m$^2$) | RMSEs(W/m$^2$) | Rs |
|---|---|---|---|---|---|
| S↑ | Noah/SLUCM | 11.98 | 12.73 | 22.86 | 0.988 |
| | IUM | 0.32 | 2.74 | 5.03 | 0.988 |
| L↑ | Noah/SLUCM | 26.18 | 26.59 | 36.58 | 0.954 |
| | IUM | 37.19 | 46.98 | 69.25 | 0.853 |
| Rn | Noah/SLUCM | -52.33 | 52.41 | 67.93 | 0.997 |
| | IUM | -31.62 | 42.57 | 62.62 | 0.988 |
| H | Noah/SLUCM | 39.20 | 44.65 | 62.13 | 0.824 |
| | IUM | 47.39 | 57.45 | 97.26 | 0.800 |
| LE | Noah/SLUCM | -33.89 | 42.82 | 88.20 | 0.199 |
| | IUM | -25.78 | 45.60 | 88.59 | 0.205 |
| G | Noah/SLUCM | -30.71 | 64.03 | 105.89 | 0.856 |
| | IUM | -80.99 | 111.37 | 169.85 | 0.552 |





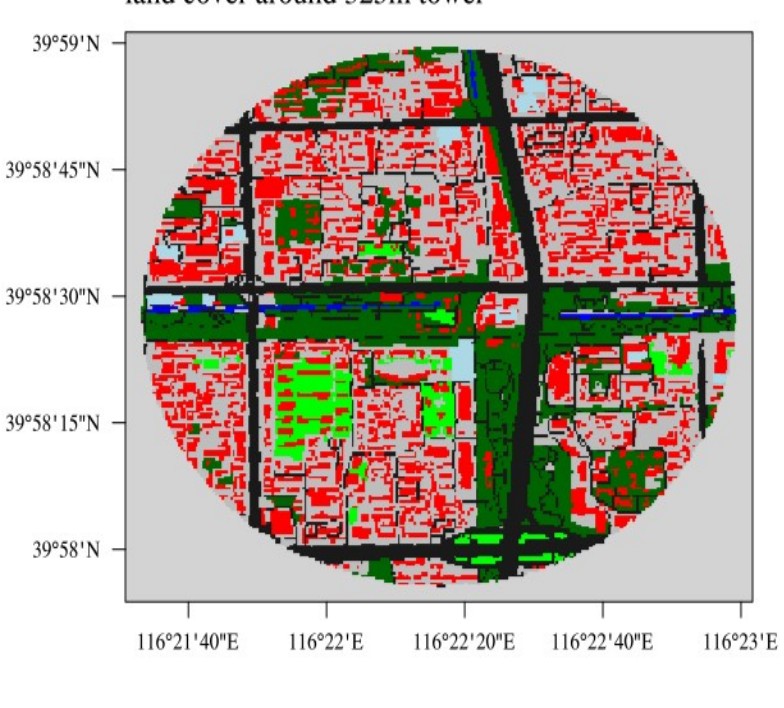


Figure 1 The LULC with 1km radius around the tower from Gaofen-2 2m resolution satellite data

1.   Trees; 2. Other impervious surfaces; 3. Grass; 4. Parking lots; 5. Roads; 6. Water; 7. Buildings









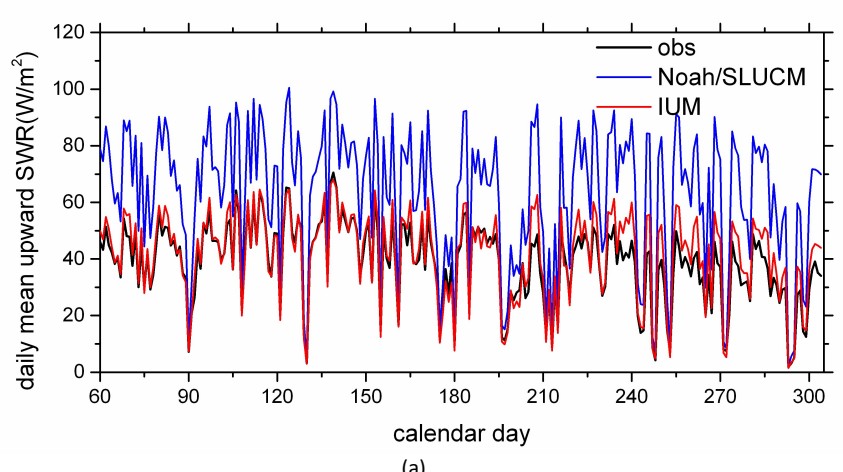

(a)

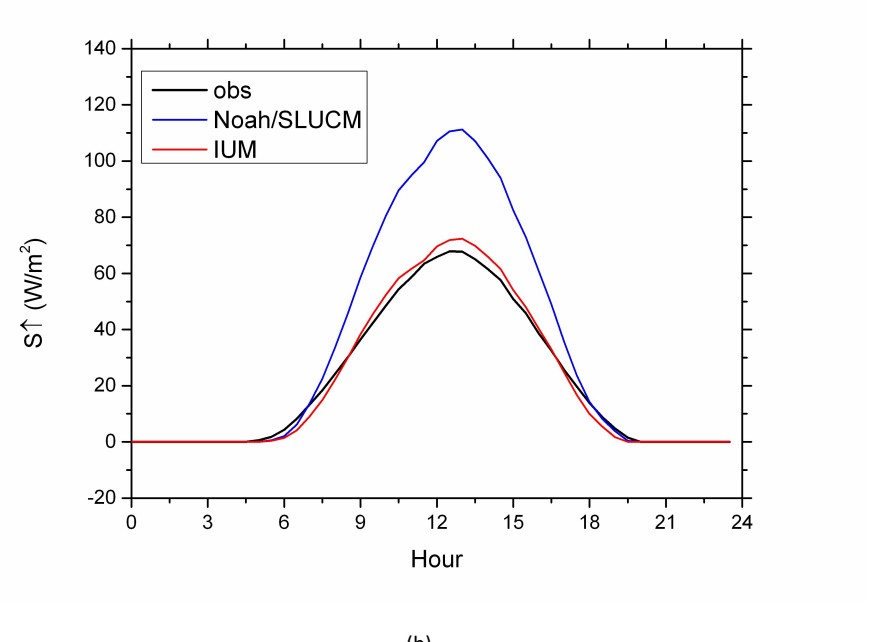


(b)

Figure 2 Daily mean (a) and diurnal cycle (b) upward shortwave radiation simulated by the

Noah/SLUCM and IUM compared with the observation.





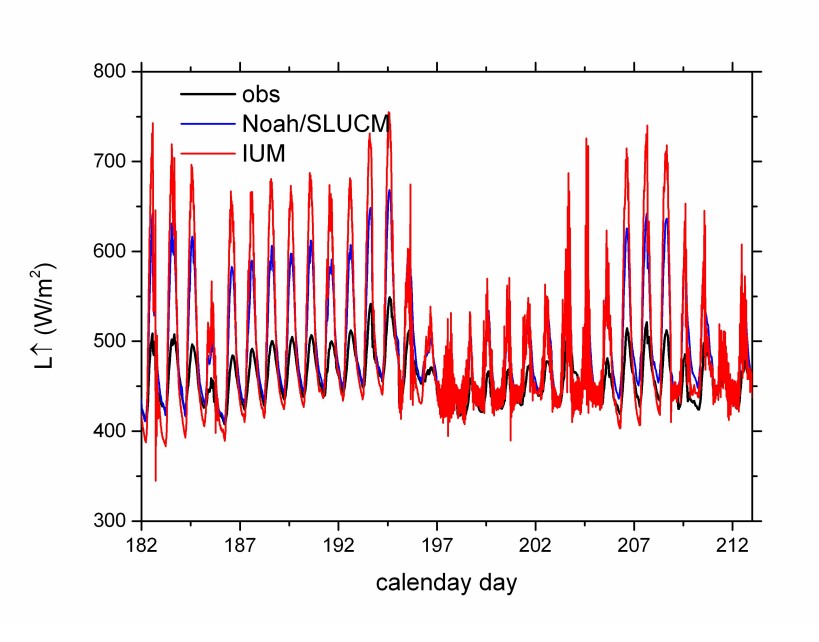


(a)

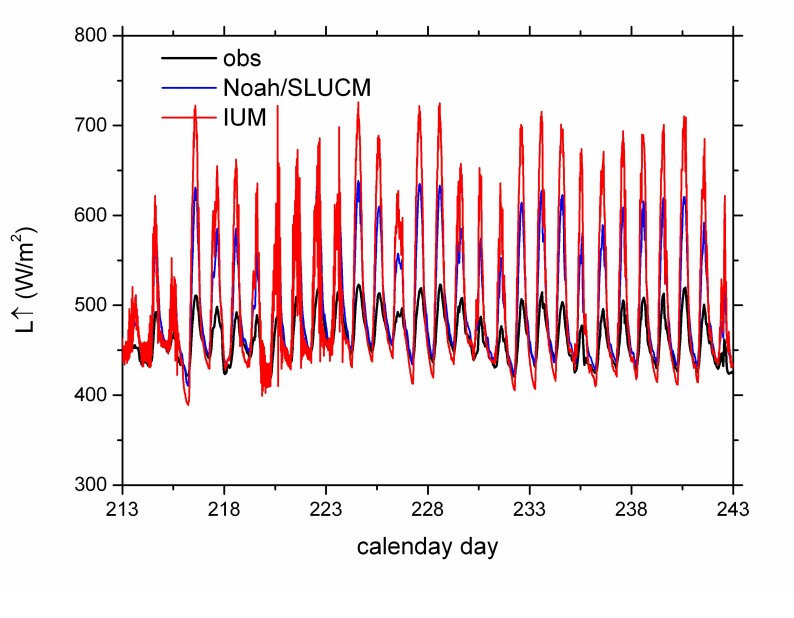


(b)



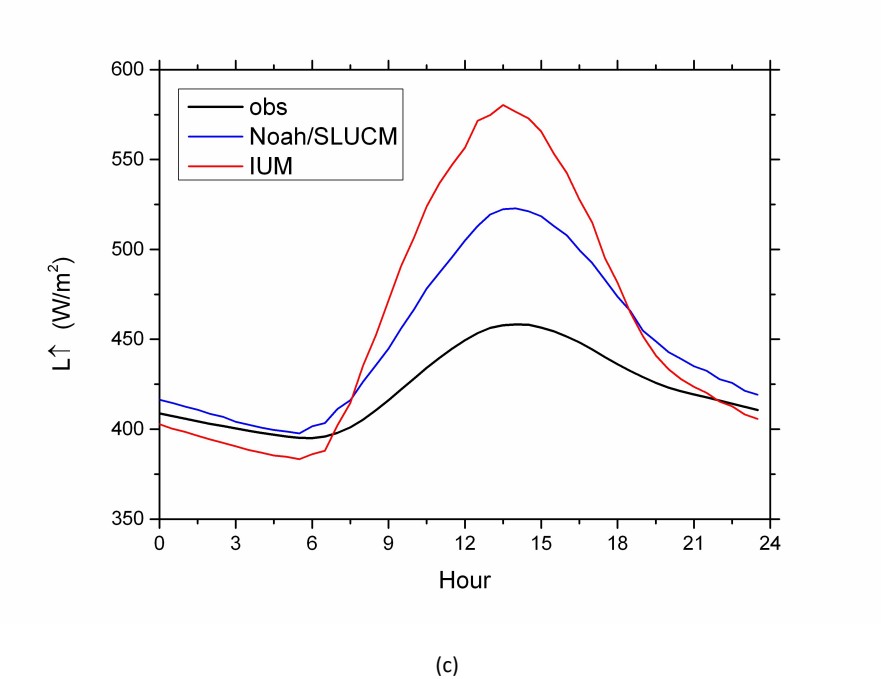


(c)

Figure 3 July (a), August (b) and diurnal cycle (c) upward longwave radiation simulated by the

Noah/SLUCM and IUM compared with the observation.








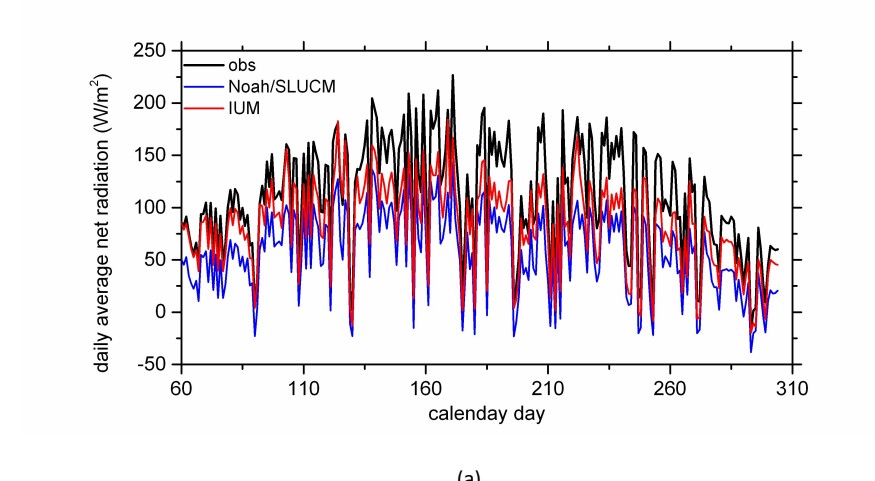


(a)

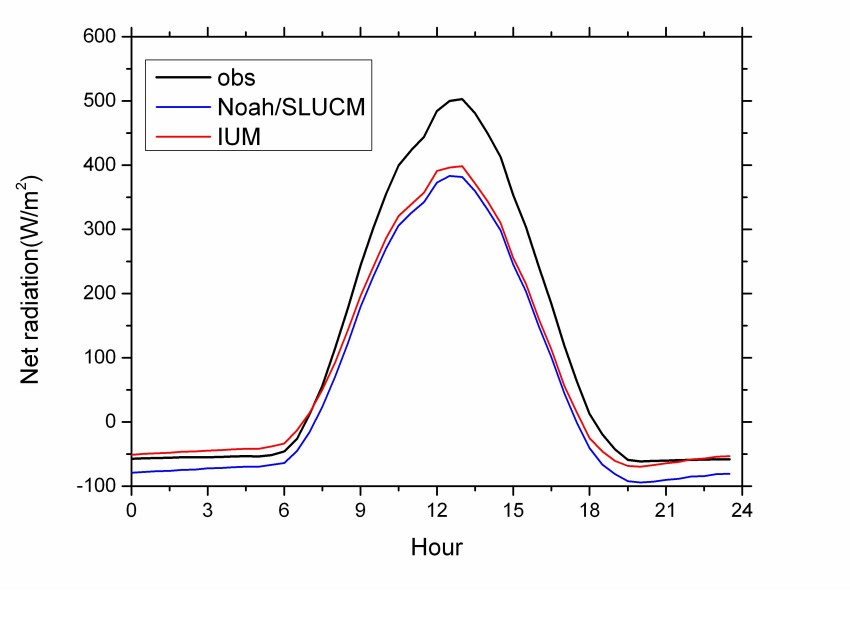


(b)

Figure 4 As figure 2, but for net radiation.








(a)


(b)

Figure 5 As figure 2, but for sensible heat flux.






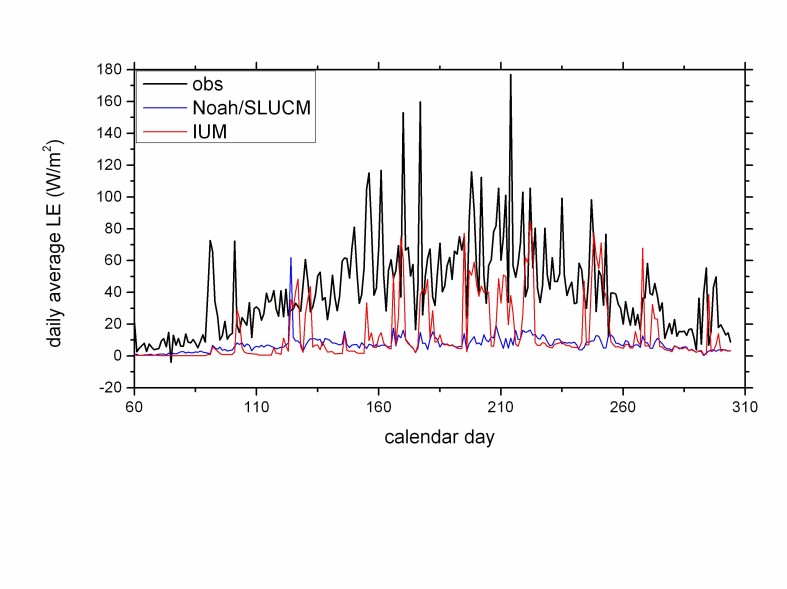

(a)

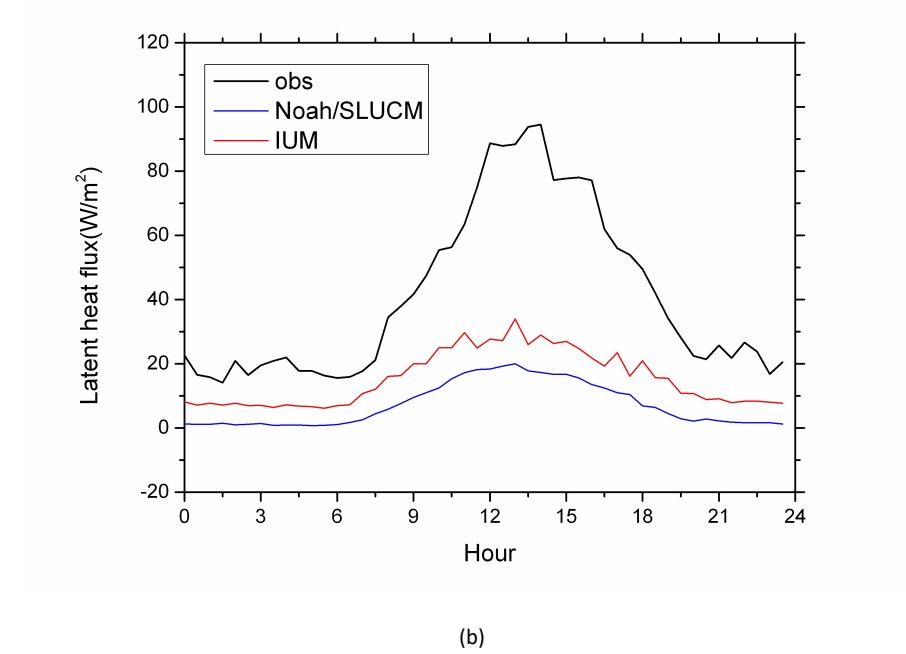


(b)

Figure 6 As figure 2, but for latent heat flux.




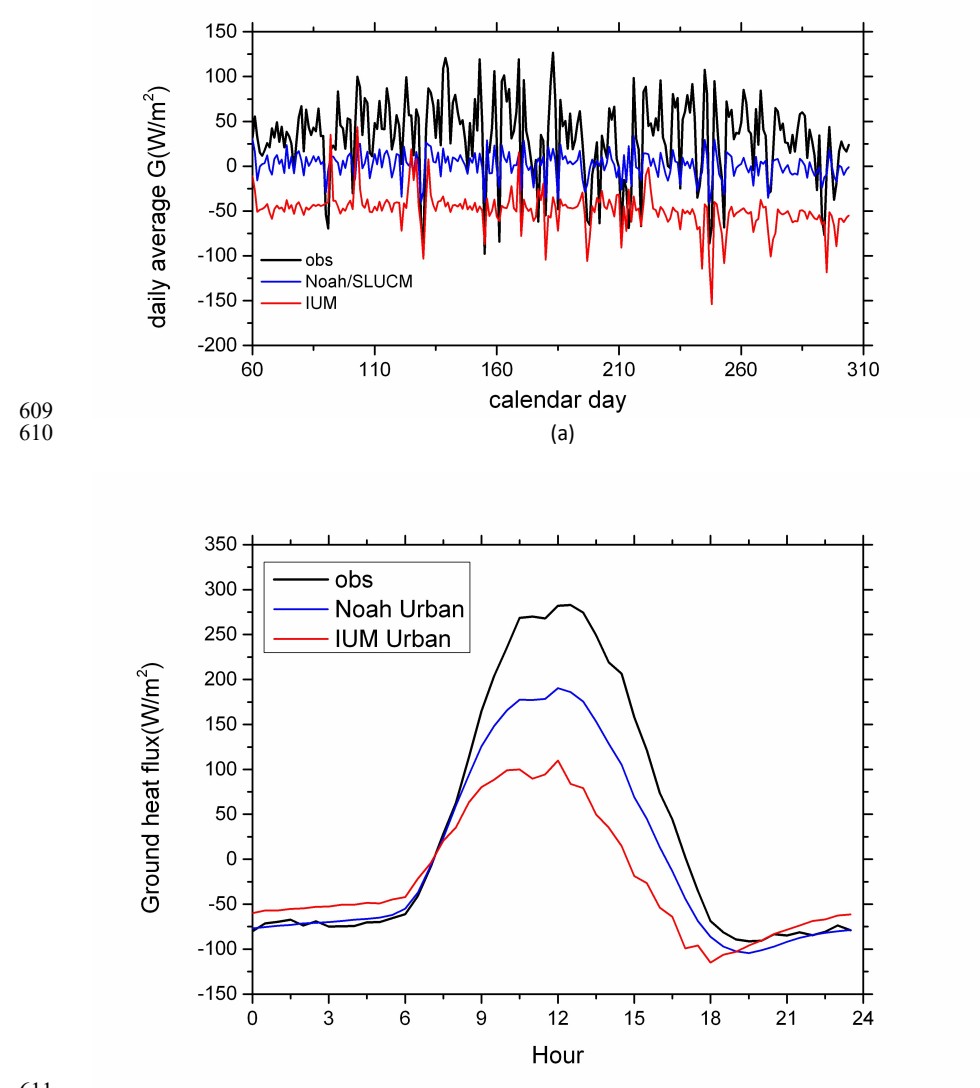

(a)


(b)

Figure 7 As figure 2, but for ground heat flux.


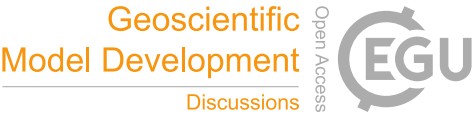

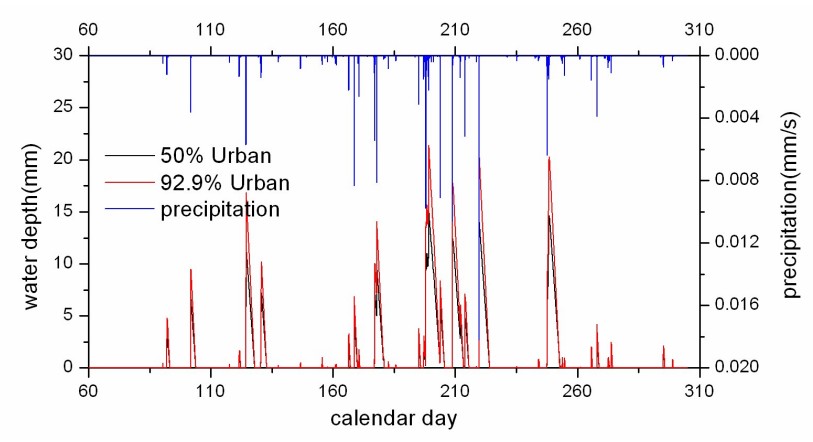


Figure 8 Precipitation and simulated water depth at two circumstances.










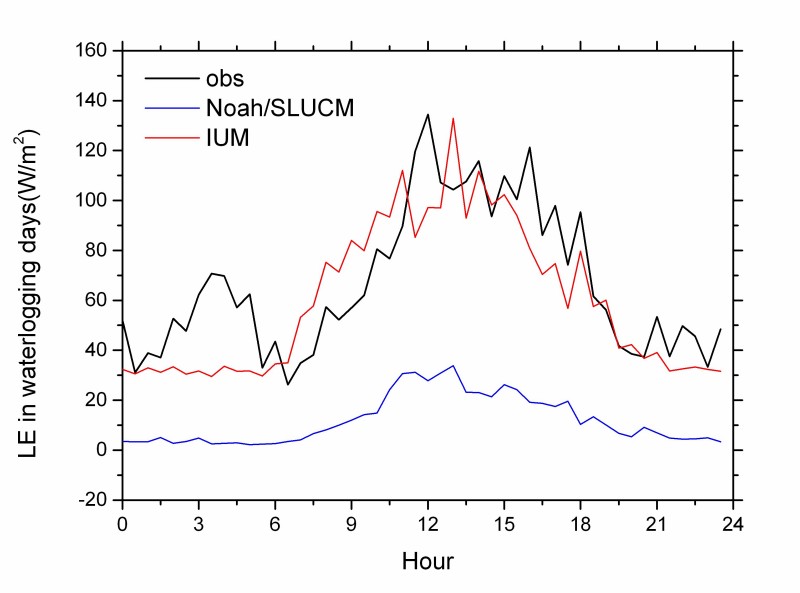

Figure 9 Observed and simulated diurnal cycle of latent heat fluxes in waterlogging days.












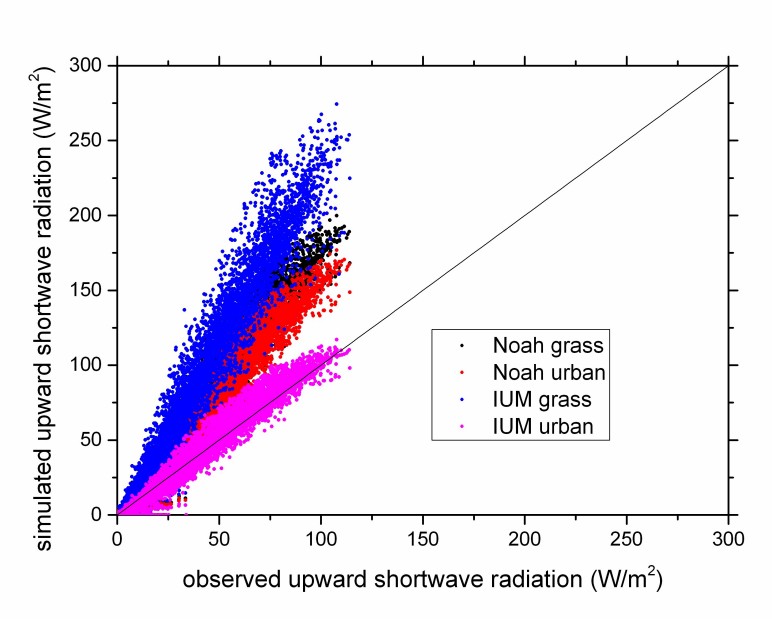


(a)


(b)



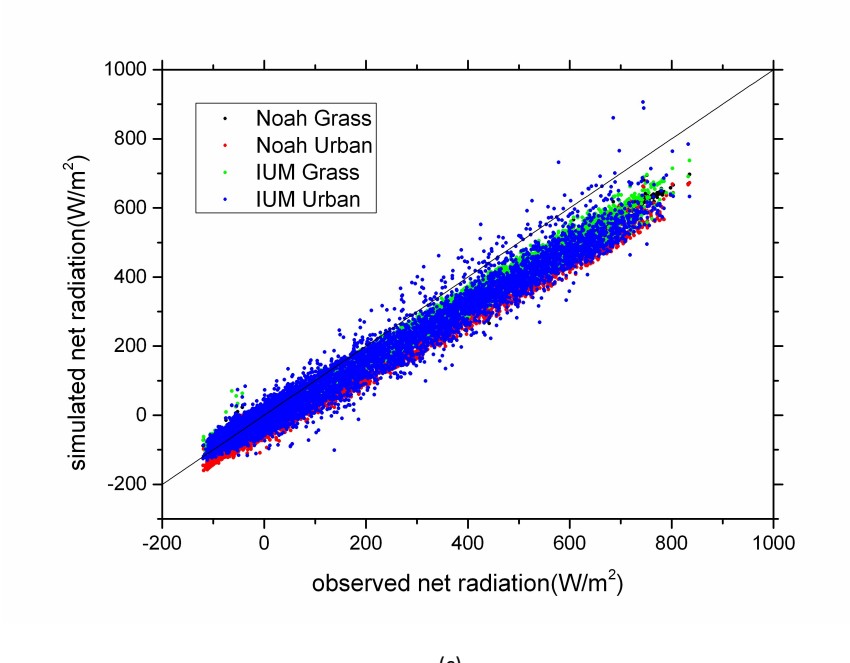


(c)

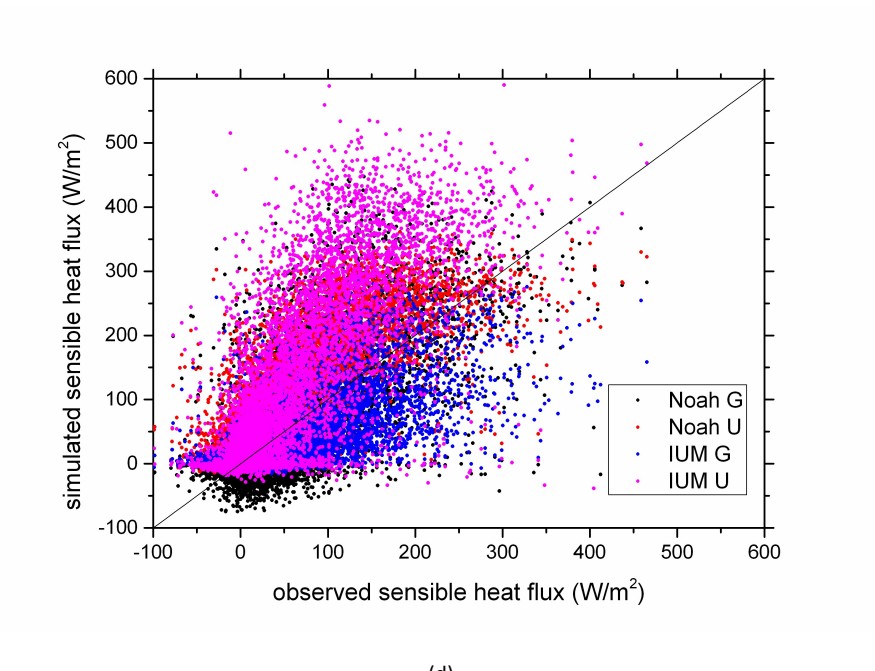


(d)



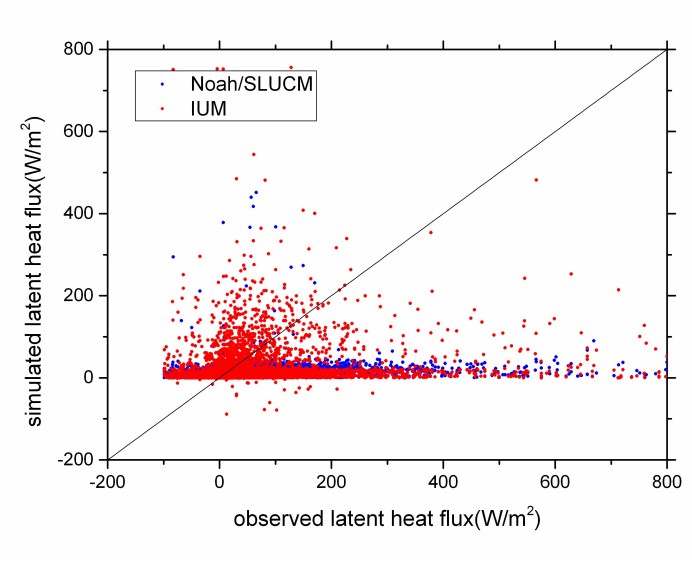


(e)

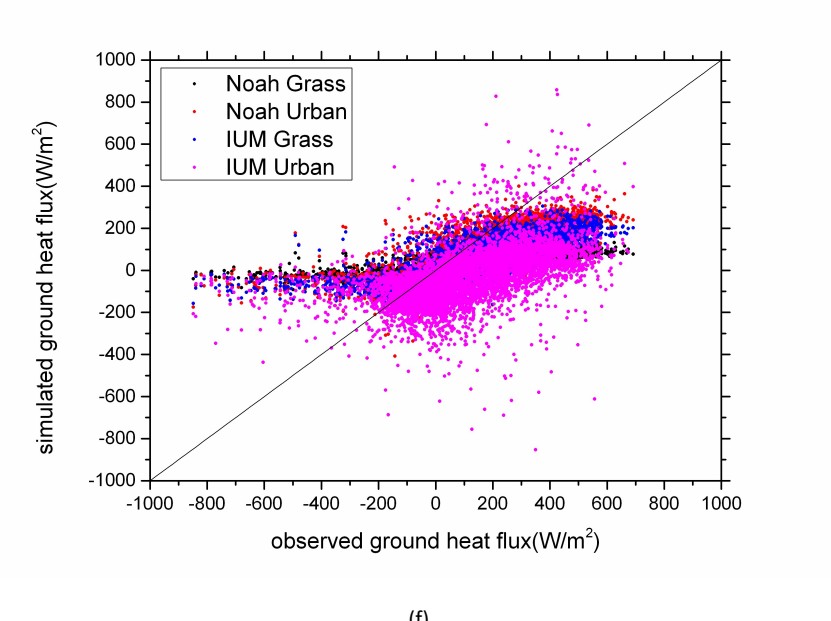


(f)

Figure 10 Scatted plots of the simulated upward shortwave radiation (a), upward longwave

radiation (b), net radiation (c), sensible heat flux (d), latent heat flux (e) and ground heat flux

(f) compared with the observation.



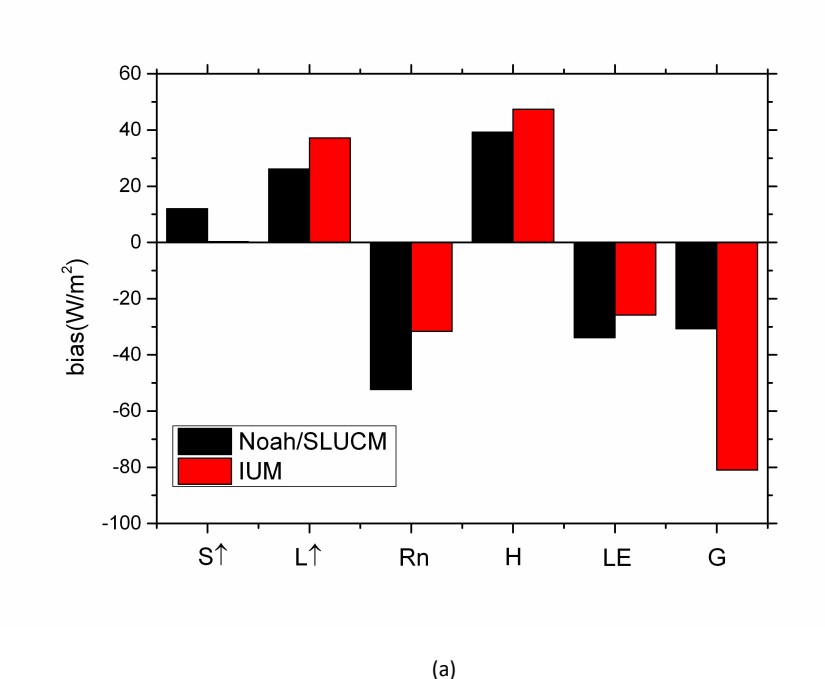


(a)

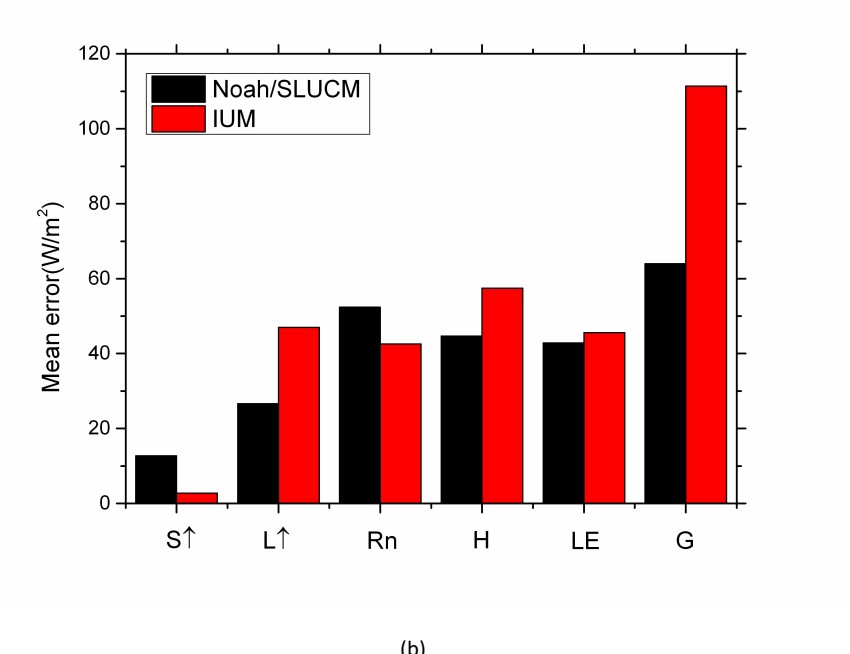


(b)





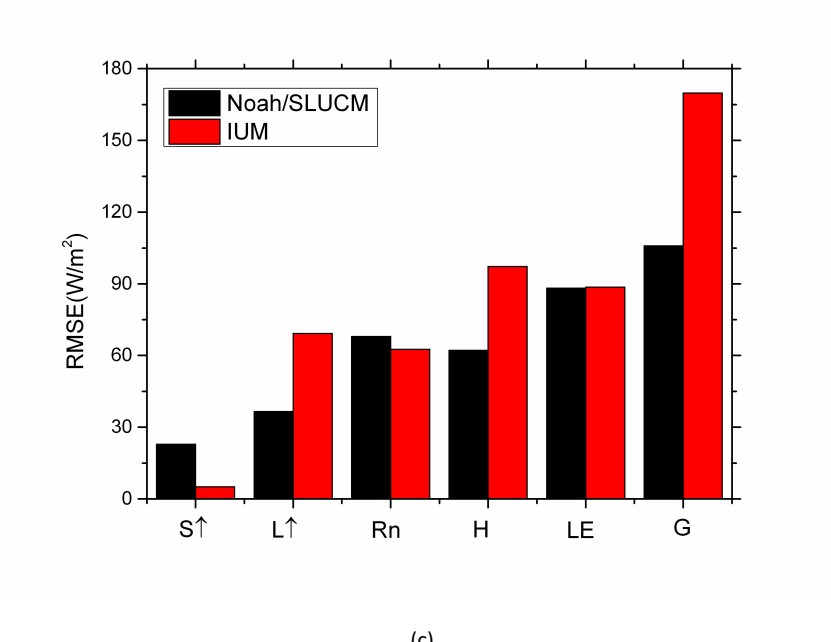


(c)

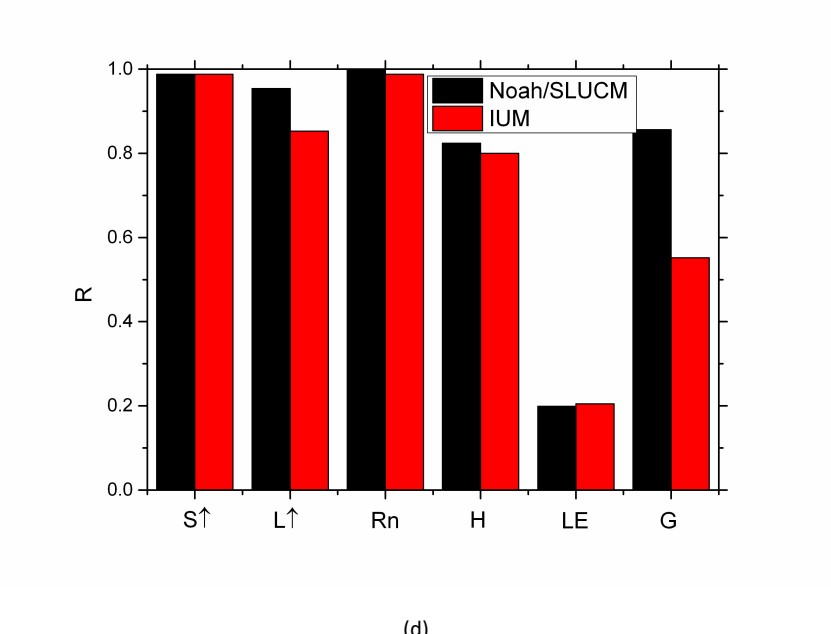


(d)

Figure 11 Biases (a), mean errors (b), root square mean errors (c) and correlation coefficients (d)

of the fluxes simulated by the two models compared with those of the observations.