# Peer review of "Intercomparison between the Integrated Urban land Model and the Noah Urban Canopy Model"

_Geoscientific Model Development, 2019_

## Referee Comment (RC1) · Anonymous Referee #1 · 11 Mar 2020

General comments: This paper intercompared the Noah Single Layer Urban Canopy Model (Noah/SLUCM) and the Integrated Urban land Model (IUM) using the observed fluxes data at the 325-meter meteorology tower in Beijing. Overall, this paper is well organized and easily to read; I suggest this paper should be published after a MINOR revision.

Specific comments: 1. Line 62, the key factors of the land surface models were found out. 2. Line 67, land surface comparison projects were launched. 3. Line 86, please give the full name of the acronym ISE. 4. Line 108, in urban areas are given below. 5. Equation 5, please give the equations for calculating the sensible heat flux from roof,

wall and road. 6. Line 193-196, road water depth should be impervious surface water depth. 7. Line 202, road surfaces should be impervious surfaces. 8. Line 223, please give the full name of the acronym LULC. 9. Line 227, the temporal resolutions of the two models are 30min. 10. Line 261, in figure 3, the amplitude of the simulation results seem too big for both the two models, maybe the heat capacity and heat conductivity should be adjusted. 11. Line 285, as the friction velocity is extremely important and associated with the roughness, the roughness in table 1 should be adjusted too reduce the bias of the sensible heat flux. 12. Figure 7, the figure (a) is not in accordance with the figure (b), please redraw either of them. 13. Equation 22, please give the meaning of Prcp and Drain. 14. Equation 22, please give the equations for calculating the roof interception, infiltration and drainage. 15. Conclusions. The urban canopy dependency parameterization (SURY) (Wouters et al., gmd-9-3027_3054, 2016) maybe a choice for urban energy balance parameterization. 16. Table 1, road should be ground.

---

## Author Comment (AC1) · 13 Mar 2020

**General comments:**

This paper intercompared the Noah Single Layer Urban Canopy Model (Noah/SLUCM) and the Integrated Urban land Model (IUM) using the observed fluxes data at the 325-meter meteorology tower in Beijing. Overall, this paper is well organized and easily to read; I suggest this paper should be published after a MINOR revision.

**Specific comments:**

1.  Line 62, the key factors of the land surface models were found out.

    **Answer**:This sentence was corrected.

2.  Line 67, land surface comparison projects were launched.

    **Answer**:This sentence was corrected.

3.  Line 86, please give the full name of the acronym ISE.

    **Answer**: impervious surface evaporation

4.  Line 108, in urban areas are given below.

    **Answer**:This sentence was corrected.

5.  Equation 5, please give the equations for calculating the sensible heat flux from roof, wall and road.

    **Answer**: These equations were added.

$H_R$, $H_B$ and $H_G$ are the sensible heat flux from roof, wall and road respectively (W m⁻²), they are calculated as follows:

$$H_R = Ch_r u_a(T_r - \theta_a) \tag{6}$$

$$H_B = Ch_b u_c(T_b - T_c) \tag{7}$$

$$H_G = Ch_g u_c(T_g - T_c) \tag{8}$$

Where $Ch_r$, $Ch_b$ and $Ch_g$ are the heat transfer coefficient from roof, wall and road respectively (mm m⁻¹); $u_a$ and $u_c$ are wind speed in the reference height and canopy respectively (m s⁻¹); $\theta_a$ is the air potential temperature at reference height (K); is the $T_c$, $T_r$, $T_b$, and $T_g$ are canopy air temperature, roof temperature, wall temperature and ground surface temperature (K), respectively.

6. Line 193-196, road water depth should be impervious surface water depth.

**Answer**:These sentences were corrected.

7. Line 202, road surfaces should be impervious surfaces.

**Answer**:This sentence was corrected.

8. Line 223, please give the full name of the acronym LULC.

**Answer:** land use and land cover

9. Line 227, the temporal resolutions of the two models are 30min.

10.**Answer**:This sentence was corrected.

11. Line 261, in figure 3, the amplitudes of the simulation results seem too big for both the two models, maybe the heat capacity and heat conductivity should be adjusted.

**Answer:** Yes, these two parameters are both important in ground surface temperature simulation. The parameterization of these parameters should be studied in the future.

12. Line 285, as the friction velocity is extremely important and associated with the roughness, the roughness in table 1 should be adjusted to reduce the bias of the sensible heat flux.

**Answer:** Yes, the roughness length of the ground is also an important parameter in sensible heat flux simulation and should be adjusted in the future too.

13. Figure 7, the figure (a) is not in accordance with the figure (b), please redraw either of them.

**Answer:** I checked figure 7 carefully, as figure 7(a) is the daily average ground heat flux, so figure 7 has no problem.

14. Equation 22, please give the meaning of Prcp and Drain.

**Answer:** $P_{rcp}$ is the precipitation (mm/s); $D_{rain}$ is the water drainage.

15. Equation 22, please give the equations for calculating the roof interception, infiltration and drainage.

**Answer: The roof rainfall interception is associated with the precipitation and the**

maximum roof interception depth, which could be calculated as follows:

$$I_{roof} = \begin{cases} 0 & W_f \geq W_{max} \\ P_{roof} P_{rcp} & W_f < W_{max} \end{cases} \qquad (26)$$

Where $P_{roof}$ is the fraction of the roof area; $W_f$ is the roof water depth; $W_{max}$ is the maximum roof interception depth. The infiltration rate is associated with the soil infiltration rate:

$$I_{nf} = min\left(\frac{P_{er}W}{\Delta t},\ I_{soil}\right) \qquad (27)$$

Where $\Delta t$ is the time step of the model; $I_{soil}$ is the infiltration rate of soil (mm/s). The water drainage is assumed equal to light to moderate rain, which is approximately 10 mm d$^{-1}$. The roof interception and the infiltration are both not considered in this paper.

16. Conclusions. The urban canopy dependency parameterization (SURY) (Wouters et al., gmd-9-3027_3054, 2016) maybe a choice for urban energy balance parameterization.

Answer: Yes, this was added as the last sentence of this section.

   The urban canopy dependency parameterization schemes (Wouters et. al, 2016) should be considered to improve the simulation associated with the urban energy balance.

17. Table 1, road should be ground.

Answer: Table 1 was corrected.

---

## Referee Comment (RC2) · Anonymous Referee #2 · 21 Mar 2020

General comments:

The study compared the results from IUM and NoahUCM against the measurements in Beijing, China. The comparison is focused on the energy fluxes (upwelling components in radiation, Rn, H, LE, and G), via the discussion on the different algorithms used in the two models. Most time-invariant parameters were retrieved from the default values or look-up tables without specific calibration. The authors drew the conclusions based on the difference in model mechanisms. It is noteworthy that, however, neither model was capable of producing reasonably accurate estimation at the study site with the setting in the current study; this amounts to the main (and major) flaw of this

manuscript. The conclusions are primarily on the suggestions for future studies and applications of the two models, including the use of remote sensing albedo in NoahUCM, considering of urban street geometry in IUM, parameterization of friction velocity, and the importance of urban hydrological models, etc. Since both NoahUCM and IUM showed significant errors on the considered energy fluxes (except for upwelling shortwave radiation), these suggestions, though applicable for urban studies in the broad sense, cannot be substantiated based on the results. The lack of model calibration against observational data in this study is hardly acceptable. It is recommended that the manuscript should be revised by including appropriate and careful calibration and evaluation of, at least one model, against quality observation and compare the other to the calibrated benchmark; or to calibrate both to the observation and compare the difference in model mechanisms. For both options, the inclusion of the observation data is essential. Overall, the results are not able to fully answer the study question raised in the Discuss stage. Both the technical merit and the quality of presentation (figures, typos, etc.) of this manuscript needs to be substantively improved before it is fitting for publication. Specific comments are detailed below.

Specific comments:

1) Detail information on LULC is needed for readers, such as the average building height, canyon aspect ratio, fractions for each LULC category, etc.

2) Much redundant but unnecessary information is provided, such as content in line 258-260, line 370-372 and line 388.

3) Many vague descriptions in quantitative comparison: such as 'too high' (line 265, 295), 'little higher' (line 273), 'little lower' (line 273), 'apparently larger' (line 292, 302), etc. It is recommended to use percentage to show the difference accurately.

4) The use of capitalization in figure labels needs to be consistent. It is recommend to capitalize all labels like Figure 4b.

[Figure]

5) In Figure 8, it is necessary to explain the different (or relation) between '50% Urban' and '92.9% Urban' in the main text.

6) Figure 9 shows in waterlogging day, IUM performs significantly better than NoahUCM in LE estimation, while both models are equally bad for non-waterlogging day (Figure 7b, Figure 10e). Does ISE only work in the event of precipitation? What caused this issue?

7) In Figure 10, it is necessary to explain the difference (or relation) between 'Grass' and 'Urban' cases. Also, figure legends need to be consistent in the subfigures, i.e. use ONE of 'Noah Grass', 'Noah G', or 'Noah grass', not all of them in different subfigures.

8) Figure 10c has different colors from all other subfigures. It is recommend to keep color consistency in subfigures as well.

9) It is recommended to show the comparison of simulated and observed ground temperature for its importance (line 265, 286, 311, 316-317, etc.).

10) Check typos and gramma. Examples: Line 273: litter -> little; X-axis labels in Figure 3a, 3b, 4a: calenday -> calendar.

---

## Author Comment (AC2) · 17 Apr 2020

General comments: The study compared the results from IUM and NoahUCM against the measurements in Beijing, China. The comparison is focused on the energy fluxes (upwelling components in radiation, Rn, H, LE, and G), via the discussion on the different algorithms used in the two models. Most time-invariant parameters were retrieved from the default values or look-up tables without specific calibration. The authors drew the conclusions based on the difference in model mechanisms. It is noteworthy that, however, neither model was capable of producing reasonably accurate estimation at the study site with the setting in the current study; this amounts to the main (and major) flaw of this manuscript. The conclusions are primarily on the suggestions for future studies and applications of the two models, including the use of remote sensing albedo in NoahUCM, considering of urban street geometry in IUM, parameterization of friction velocity, and the importance of urban hydrological models, etc. Since both NoahUCM and IUM showed significant errors on the considered energy fluxes (except for upwelling shortwave radiation), these suggestions, though applicable for urban studies in the broad sense, cannot be substantiated based on the results. The lack of model calibration against observational data in this study is hardly acceptable. It is recommended that the manuscript should be revised by including appropriate and careful calibration and evaluation of, at least one model, against quality observation and compare the other to the calibrated benchmark; or to calibrate both to the observation and compare the difference in model mechanisms. For both options, the inclusion of the observation data is essential. Overall, the results are not able to fully answer the study question raised in the Discuss stage. Both the technical merit and the quality of presentation (figures, typos, etc.) of this manuscript needs to be substantively improved before it is fitting for publication.

Answer: Model calibration was implemented for IUM. Firstly, we chose some key parameters in the urban radiation balance model and energy balance model. Then we adjusted the values of these parameters to calibrate the IUM. The criterion for the calibration is the minimum of the aggregate root mean square error and bias of the six radiative and energy fluxes, i. e. upward solar radiation, upward longwave radiation, net radiation, sensible heat flux, latent heat flux and ground heat flux. Lastly, we used some of these parameters to Noah/SLUCM. Apparently, the key parameters in urban radiation balance model are albedos for visible and near infrared solar radiation. The key parameters selected in the urban energy balance model include surface thermal capacity, surface thermal conductivity, bulk urban roughness and number of impervious layers.

Specific comments are detailed below. Specific comments: 1) Detail information on

LULC is needed for readers, such as the average building height, canyon aspect ratio, fractions for each LULC category, etc. Answer: The average building height is 40m. The canyon height-to-width ratio is approximately 1. The fractions for trees, other impervious surfaces, grass, parking lots, roads, water, and buildings are 0.1600, 0.3538, 0.0353, 0.0153, 0.1930, 0.0063 and 0.2363, respectively.

2) Much redundant but unnecessary information is provided, such as content in line 258-260, line 370-372 and line 388. Answer: These sentences were deleted.

3) Many vague descriptions in quantitative comparison: such as 'too high' (line 265, 295), 'little higher' (line 273), 'little lower' (line 273), 'apparently larger' (line 292, 302), etc. It is recommended to use percentage to show the difference accurately. Answer: These vague descriptions were replaced by quantitative percentage comparison.

4) The use of capitalization in figure labels needs to be consistent. It is recommend to capitalize all labels like Figure 4b. Answer: The capitalization in figure labels were corrected.

5) In Figure 8, it is necessary to explain the different (or relation) between '50% Urban' and '92.9% Urban' in the main text. Answer: This figure was redrawn.

6) Figure 9 shows in waterlogging day, IUM performs significantly better than NoahUCM in LE estimation, while both models are equally bad for non-waterlogging day (Figure 7b, Figure 10e). Does ISE only work in the event of precipitation? What caused this issue? Answer: ISE works in the event of waterlogging, not in the event of precipitation. For dry impervious surface days, the ISE is zero. The simulations for the LE are both these two models are bad when the impervious surface is dry because the LE simulation for pervious surface is still need to be improved.

7) In Figure 10, it is necessary to explain the difference (or relation) between 'Grass' and 'Urban' cases. Also, figure legends need to be consistent in the subfigures, i.e. use ONE of 'Noah Grass', 'Noah G', or 'Noah grass', not all of them in different subfigures.

Answer: This figure was redrawn.

8) Figure 10c has different colors from all other subfigures. It is recommend to keep color consistency in subfigures as well. Answer: This figure was redrawn.

9) It is recommended to show the comparison of simulated and observed ground temperature for its importance (line 265, 286, 311, 316-317, etc.). Answer: Ground temperature is important to the simulation of sensible heat flux. But unfortunately, no observed ground temperature is available. We compared the simulated ground temperature of these two models.

10) Check typos and gramma. Examples: Line 273: litter -> little; X-axis labels in Figure 3a, 3b, 4a: calenday -> calendar. Answer: The typos and gramma were checked carefully.

---

## Author Comment (AC3) · 21 Apr 2020

As the IUM model was calibrated, some figures were redrawn as the supplement.

Please also note the supplement to this comment:
https://www.geosci-model-dev-discuss.net/gmd-2019-298/gmd-2019-298-AC3-supplement.zip